# Integrative Multiomics Profiling Unveils the Protective Function of Ulinastatin against Dextran Sulfate Sodium-Induced Colitis

**DOI:** 10.3390/antiox13020214

**Published:** 2024-02-08

**Authors:** Tianyu Yu, Jun Yan, Ruochen Wang, Lei Zhang, Xiake Hu, Jiaxi Xu, Fanni Li, Qi Sun

**Affiliations:** 1Department of General Surgery, The First Affiliated Hospital of Xi’an Jiaotong University, Xi’an 710061, China; yuty2017@stu.xjtu.edu.cn (T.Y.); yanj2018@stu.xjtu.edu.cn (J.Y.); xajdzl@126.com (L.Z.); huxiake2023@163.com (X.H.); 2Center for Gut Microbiome Research, Med-X Institute, The First Affiliated Hospital of Xi’an Jiaotong University, Xi’an 710061, China; wangruochen0628@outlook.com; 3Department of Physiology and Pathophysiology, Xi’an Jiaotong University Health Science Center, Xi’an 710061, China; xujiaxi@xjtu.edu.cn; 4Department of Talent Highland, The First Affiliated Hospital of Xi’an Jiaotong University, Xi’an 710061, China

**Keywords:** colitis, ulinastatin, inflammation, microbiota, metabolite

## Abstract

Ulcerative colitis is an inflammatory bowel disease with multiple pathogeneses. Here, we aimed to study the therapeutic role of ulinastatin (UTI), an anti-inflammatory bioagent, and its associated mechanisms in treating colitis. Dextran sulfate sodium was administrated to induce colitis in mice, and a subgroup of colitis mice was treated with UTI. The gut barrier defect and inflammatory manifestations of colitis were determined via histological and molecular experiments. In addition, transcriptomics, metagenomics, and metabolomics were employed to explore the possible mechanisms underlying the effects of UTI. We found that UTI significantly alleviated the inflammatory manifestations and intestinal barrier damage in the mice with colitis. Transcriptome sequencing revealed a correlation between the UTI treatment and JAK-STAT signaling pathway. UTI up-regulated the expression of SOCS1, which subsequently inhibited the phosphorylation of JAK2 and STAT3, thus limiting the action of inflammatory mediators. In addition, 16S rRNA sequencing illustrated that UTI maintained a more stable intestinal flora, protecting the gut from dysbiosis in colitis. Moreover, metabolomics analysis demonstrated that UTI indeed facilitated the production of some bile acids and short-chain fatty acids, which supported intestinal homeostasis. Our data provide evidence that UTI is effective in treating colitis and support the potential use of UTI treatment for patients with ulcerative colitis.

## 1. Introduction

Ulcerative colitis (UC) is a multifactorial, relapsing inflammatory bowel disease (IBD) characterized by shallow ulcers in the mucosal layer of colon [1]. In 2023, it affected 5 million people worldwide, and this number is still climbing [2]. Multiple aspects of environmental exposures, like lifestyle and dietary factors, together with predisposition due to a person’s genetic background have been associated with the onset of UC, through an interplay of gut epithelial defects, the microbiota, and immune disorders [3,4,5]. UC often induces repeated abdominal pain, bloody diarrhea, and extraintestinal manifestations in some cases [6]. Notably, the function of the intestinal barrier is a primary prognostic indicator in patients with UC, highlighting that disruption of gut barrier is one of initial pathogenesis. Therefore, how to maintain mucosal homeostasis and protect the integrity of gut barrier are key in the treatment of UC. For patients with mild-to-moderate UC, 5-aminosalicylate (5-ASA) is considered the first-line option for treatment, mainly mitigating gut inflammation with poor outcomes for mucosal barrier repair. In addition, biological agents, such as TNF-α inhibitors, integrin inhibitors, and interleukin inhibitors, have been used clinically for treating UC [7]. Although the therapeutic options for UC are increasing, over 10% of patients still inevitably need to face the fate of proctocolectomy [2]. A combination of therapeutics or drugs with combined therapeutical effects is required.

The gut microbiota has gained attention as an abundant and stable ecosystem that serves to enhance the barrier function and maintain gut homeostasis. Moreover, microbial dysbiosis has been linked to the onset of UC, while the restoration of a healthy microbiome via fecal microbiota transplantation exhibits beneficial effects in the treatment of UC [8]. The interaction between the microbes and hosts is achieved primarily through metabolites, especially bile acids (BAs) and short-chain fatty acids (SCFAs), which can be affected by the diversity and composition of the gut microbiota [9]. Indeed, various bacterial metabolites, rather than the microbiota itself, are crucial to the maintenance of mucosal integrity [10]. Metabolomics studies on patients with UC have reported a decrease in the production of secondary BAs and SCFAs, suggesting that secondary BA- and SCFA-producing bacteria are depleted in UC-related microbial dysbiosis [11,12]. Thus, these metabolites exert barrier-protective actions and anti-inflammatory functions in the gut. However, treating UC or other IBDs with fecal microbiota transplantation is still under exploration due to its complicity and existing biosafety issues [13]. Besides the gut microbiota, a wide range of cytokines are involved in the pathogenesis of UC, and targeting cytokine-associated signaling, especially the JAK-STAT pathway, has already become a newly confirmed therapeutic strategy for UC [14,15]. Janus kinase (JAK) molecules, including JAK1, JAK2, JAK3, and tyrosine kinase (TYK) 2, are constitutively bound to cytokine receptors [16,17]. As the cytokines bind to their receptors, the JAKs and subsequent STATs are activated to trigger specific biological responses. By targeting a wide range of JAK-dependent cytokines, reports on the clinical trials of Tofacitinib, a pan-JAK inhibitor, showed efficacy in treating patients with moderate-to-severe UC [18,19]. Although JAK inhibition has exhibited therapeutic effectiveness for UC, the intricacies of the relationship between the JAK-STAT pathway and cytokines in the progression of UC remain elusive and paradoxical.

Ulinastatin (UTI), a urinary trypsin inhibitor, has shown a wide range of clinical applications through its pharmacological effects, including eliminating oxygen free radicals, lessening the excessive release of inflammatory mediators, and stabilizing the lysosomal membrane [20]. Due to its anti-inflammatory role, UTI has been used as an effective therapeutical treatment for a variety of inflammatory diseases, such as acute and chronic pancreatitis, pneumonia, acute kidney injury, and sepsis [21,22]. In addition, UTI has also exerted protective effects on the intestinal mucosal barrier function in rats with circulatory failure and organ dysfunction [23]. Therefore, UTI might alleviate the pathologic symptoms of UC by improving the colonic mucosal barrier and inhibiting cytokine-associated inflammatory signaling. To test this hypothesis, we determined the therapeutical effect of UTI in a mouse colitis model induced by dextran sulfate sodium (DSS) and elucidated the potential mechanisms of UTI on the composition of gut microbiota and specific anti-inflammatory effects on UC.

Here, we found that the treatment with UTI significantly alleviated the inflammatory manifestations and intestinal barrier damage in mice with colitis. The transcriptome sequencing and molecular experiments revealed a correlation between the UTI treatment and JAK2/STAT3/SOCS1 signaling pathway. Meanwhile, 16S rRNA sequencing showed that gut microbial dysbiosis was corrected by UTI. In particular, UTI increased the abundance of beneficial *Lactobacillus*, while removing the pathogenic bacteria. Metabolomics analysis showed that UTI facilitated the production of certain BAs and SCFAs, which are essential for intestinal homeostasis. Together, these findings demonstrate the potential of UTI as a therapeutic treatment for UC and provide a candidate drug with combined effects of anti-inflammation and gut microbiota protection.

## 2. Materials and Methods

### 2.1. Chemicals and Reagents

DSS (molecular weight 36,000–50,000) was purchased from MP Biomedicals (Santa Ana, CA, USA). UTI (100,000 U) was purchased from Techpool Biochemical Pharmaceutical Co., Ltd. (Guangzhou, China). 5-ASA was purchased from Meilunbio Biotech Co., Ltd. (Dalian, China). A luminol chemiluminescence probe L-012 was obtained from Wako Biomedicals (Osaka, Japan). Fluorescein isothiocyanate-dextran (4 kDa) (FD4) was purchased from Yuanye BioTech (Shanghai, China). Assay kits of reactive oxygen species (ROS), malondialdehyde (MDA), myeloperoxidase (MPO), and superoxide dismutase (SOD) were purchased from Nanjing Jiancheng Bioengineering Institute (Nanjing, China). All enzyme-linked immunosorbent assay (ELISA) kits were purchased from Thermo Fisher Scientific lnc. (Waltham, MA, USA). An RNA isolation kit was purchased from Promega Biotech Co., Ltd. (Fitchburg, WI, USA). All protein extraction and quantification reagents were purchased from Solarbio Science & Technology Co., Ltd. (Beijing, China).

### 2.2. Animals and Colitis Model Establishment

C57BL/6J mice (female; 6 weeks of age; weight 18–20 g) were purchased from the Experimental Animal Center of Xi’an Jiaotong University. The animals were raised under pathogen-free conditions (ambient temperature of 25 ± 2 °C; relatively constant humidity of 50 ± 10%; 12 h light/dark cycle) and had access to water and food ad libitum.

Prior to DSS induction, all the mice experienced 1 week of acclimatization. The mice were distributed randomly among four groups, with 8 mice per group: control group, acute colitis group, UTI-treated group, and 5-ASA positive control group. Acute colitis was induced by adding 3% DSS to the drinking water for 7 days. A daily intraperitoneal injection of UTI (10,000 U/kg) was initiated and ended consistently with DSS induction in the UTI group. 5-ASA, the first-line anti-inflammatory therapy for UC, was used as a positive control to evaluate the efficacy of UTI. Similarly, the 5-ASA group received a daily intragastric administration of 5-ASA (200 mg/kg). The symptoms of the mice were monitored daily, and the disease activity index (DAI) for each animal was calculated by means of body weight, stool consistency, and fecal occult blood.

### 2.3. Specimen Collection and Evaluation of Colitis

On the ninth day after the initiation of treatment, the mice were sacrificed under anesthesia, and their blood, tissues, and colonic contents were collected, processed, and stored. Specifically, the whole colons were removed, gently rinsed with cold phosphate-buffered saline (PBS), and the fat and mesentery were clipped. The length of each colon was measured, and then cut off to collect the colonic contents. The entire process of mice dissection and sample collection was performed on a vertical flow clean bench to avoid bacterial contamination. The distal colon specimens were fixed in 4% paraformaldehyde overnight for histological, immunohistochemistry, and immunofluorescence assessments. The remaining colons were subsequently divided into fragments, snap-frozen in liquid nitrogen, and kept at −80 °C until used for biochemical determination, RNA isolation, and protein extraction. The spleens were removed and weighted to calculate the spleen index (spleen/body weight ratio).

### 2.4. Histopathological Analysis

After fixation, the distal colon specimens were embedded in paraffin and sliced into 5 μm sections for hematoxylin and eosin (H&E) and alcian blue/periodic acid–Schiff (AB/PAS) staining. Colon morphology was obtained using a light microscopy. The extent of tissue damage was scored according to the degrees of inflammatory cell infiltration, epithelial changes, and ulcerative area by two experienced pathologists.

### 2.5. In Vivo Imaging of ROS

The luminol chemiluminescence probe L-012 serves as a simple and effective testing tool to monitor ROS via in vivo imaging [24]. L-012 was dissolved in PBS at a concentration of 2 mg/mL right before the experiment. The mice on the seventh day of DSS induction were anesthetized with 2.0% isoflurane, and then intraperitoneally injected with fresh L-012 (20 mg/kg). The animals were subsequently placed into the chamber of the In vivo Imaging System (IVIS) to obtain bioluminescent images.

### 2.6. Biochemical Analysis

Fresh colon tissues were separated into single-cell suspensions to determine the ROS content using detection kits. Frozen colon specimens were rapidly weighed and homogenized to prepare 10% tissue homogenate on ice. The contents of oxidation markers in tissue homogenate, including MDA, MPO, and SOD, were measured using specific assay kits according to the manufacturer’s instructions. The whole blood was centrifuged for the collection of the serum component, in which the concentration of cytokines (TNF-α, IL-1β, and IL-6) was measured with corresponding ELISA kits.

### 2.7. Intestinal Barrier Function Detection

FD4 was used to assess gut permeability as described previously [25,26]. On the last day (day 9), after being deprived of food and water for 4 h, the experimental mice received FD4 intragastric administration (600 mg/kg). Then, sera were collected 4 h later to measure the fluorescence intensity at 485/525 nm excitation/emission wavelengths. Finally, the serum FD4 concentration was calculated according to a standard curve established by a gradient dilution of FD4.

### 2.8. Immunohistochemistry and Immunofluorescence

Paraffin-embedded colon sections were dewaxed, hydrated, subjected to antigen retrieval, and blocked. For immunohistochemistry, the sections were incubated with primary antibodies, followed by horseradish peroxidase-conjugated secondary antibodies. For immunofluorescence, the sections were incubated with primary antibodies overnight at 4 °C, and subsequently fluorescein-conjugated secondary antibodies for 2 h away from the light. The sections were counterstained with DAPI and visualized using a laser confocal microscope. Information on the antibodies is listed in Appendix A.

### 2.9. Quantitative Real-Time PCR

Total RNA was extracted by grinding frozen colon tissues with an RNA isolation kit. Extracted RNA was then reverse-transcribed to complementary DNA, followed by RT-qPCR using SYBR Green PCR Master Mix in the CFX96 Real-Time System (Bio-Rad, Hercules, CA, USA). The relative expression of the target genes was normalized to the GAPDH level and calculated using the 2^−ΔΔCt^ method. The sequences of primers are listed in Appendix A.

### 2.10. Western Blotting Analysis

Total protein was extracted by grinding frozen colon tissues with RIPA lysis buffer containing 1% protease and phosphatase inhibitors, which was subsequently quantitated using a bicinchoninic acid protein assay kit. The extracted protein was diluted with sodium dodecyl sulfate-polyacrylamide gel (SDS-PAGE) loading buffer and heated to denaturation in a metal bath. Proteins of different molecular weights were separated on SDS-PAGE (10%) by electrophoresis, and subsequently transferred onto polyvinylidene difluoride membranes. After being blocked by 5% skim milk for 2 h, the membranes were incubated with primary antibodies at 4 °C overnight. After being washed in Tris-buffered saline containing 0.1% Tween20, the membranes were incubated with horseradish peroxidase-conjugated secondary antibodies for 2 h. Finally, images of protein bands were captured by a Bio-Rad Chemidoc system (Bio-Rad, Hercules, CA, USA) and quantitated with the software ImageJ (Version 1.51). Information on the antibodies is listed in Appendix A.

### 2.11. RNA Sequencing

Colon tissues were collected and kept in RNA stabilization solution. The samples were ground to extract total RNA using an RNA isolation kit. RNA integrity was assessed with an Agilent Bioanalyzer 2100 Bioanalyzer (Agilent Technologies, Santa Clara, CA, USA). An RNA integrity number > 8.0 and optical density ratios of 260/280 nm from 1.9 to 2.1 were used as the threshold for samples to construct RNA-Seq libraries. The sequencing of libraries was performed on an Illumina HiSeq platform by Sangon Biotech (Shanghai, China). The quality of the sequencing was determined using FastQC. Differential expression analysis was performed with DESeq2. The cut-off for determining the differentially expressed genes (DEGs) were a fold change of >1 and false discovery rate of <0.05. Kyoto Encyclopedia of Genes and Genomes (KEGG) pathway enrichment analysis was then employed for all the DEGs. Q  ≤  0.05 was set as the criterion for significant enrichment of KEGG pathways. 

### 2.12. 16S rDNA Gene High-Throughput Sequencing

The colon contents of mice were harvested under sterile conditions and kept at −80 °C. The samples were sent to Tsingke Biotech (Beijing, China) to extract DNA and perform high-throughput 16S rDNA sequencing using the Illumina Novaseq platform (Illumina, San Diego, CA, USA). Raw data were quality filtered using Trimmomatic (Version 0.33) and Cutadapt (Version 1.9.1). Chimeric sequences were identified and filtered using Quantitative Insights into Microbial Ecology 2 (QIME2, Version 2020.4). Non-chimeric sequences were clustered into observed taxonomic units (OTUs) at a similarity threshold of 97% using the UPARSE algorithm. The OTUs were taxonomically annotated using SILVA. Alpha and beta diversity indexes were calculated using QIIME2. Principal coordinates analysis (PCoA) was used to visualize the compositional changes in microbial community between the experimental samples based on the Bray–Curtis distance with R programming language (Version 4.1.0).

### 2.13. Intestinal Metabolomics Analysis

The colon contents samples were delivered to Metware Biotech (Wuhan, China) to employ a widely targeted metabolomics assay. The BA contents were detected using the AB Sciex QTRAP 6500 LC-MS/MS platform. A total of 20 mg of sample was extracted with methanol after grinding, and then added to the internal standard for quantitation. After protein precipitation and centrifugation, the supernatant was collected. The extracts were evaporated to dryness and reconstituted in methanol, followed by liquid chromatography–mass spectrometry analysis. Analyst (Version 1.6.3) was utilized for data collection, and Multiquant (Version 3.0.3) was utilized for metabolite quantification.

The SCFA contents were detected using the Agilent 7890B-7000D GC-MS/MS platform. A total of 20 mg of sample was weighed, dissolved in phosphoric acid solution, homogenized under ultrasonication, and centrifuged to acquire a supernatant. Following being added with tert-butyl methyl ether containing internal standard, the mixture was vortexed, ultrasonicated, and centrifuged again to collect the supernatant for gas chromatography–mass spectrometry analysis.

Orthogonal partial least squares discriminant analysis (OPLS-DA) was performed to analyze the data to explore the differences in metabolic profiles between each group.

### 2.14. Statistical Analysis

Statistical analysis was carried out using GraphPad Prism (Version 9.5.1) and SPSS (Version 29.0). Values are presented as mean ± standard error mean (SEM). Multiple group comparison was performed using one-way or two-way analysis of variance (ANOVA), followed by Tukey’s multiple comparison test. If the data did not fulfill the prerequisites of parametric statistics, a Kruskal–Wallis test followed by Dunn’s multiple comparisons was performed. *p* < 0.05 was considered as statistically significant.

## 3. Results

### 3.1. UTI Administration Ameliorated Pathological Features in DSS-Induced Colitis

To investigate the therapeutic efficacy of UTI against IBD, models of mice with acute colitis induced by DSS were established. The mice were acclimatized for 1 week and subsequently received 3% DSS in the drinking water for 7 days, during which the mice were treated with UTI (10,000 U/kg/d) or a vehicle, weighed daily (at a similar time of day), and monitored for clinical symptoms (Figure 1). The administration dosage of UTI was determined and modified based on the prior reports of other inflammatory disease models [27,28]. The assessment of colitis severity typically is focused on body-weight loss, diarrhea, and hematochezia, which were quantified using the DAI. The DSS-induced mice developed colitis symptoms on day 5, followed by a gradual increase in the DAI (Figure 2A,B). Although colitis symptoms appeared in all the DSS-induced mice, the weight loss and DAI of the UTI and 5-ASA groups were significantly lower than those of the colitis group. Furthermore, for anatomic observation, colonic and systematic inflammation were characterized by shortened colons and enlarged spleens, which emerged in the DSS-induced groups (Figure 2C). Compared with the colitis group, the UTI treatment markedly relieved colonic shortening and improved the spleen index (spleen/body weight ratio) (Figure 2D,E). The histopathological examination of colonic sections illustrated that DSS induction triggered mucosal damage, ulceration, inflammatory cell infiltration, and crypt destruction. Especially in the colitis group, the pathological sections exhibited apparent mucosal erosions and leukocyte infiltration (Figure 2F). All of the above-described symptoms and pathologic manifestations point to the successful construction of this colitis model. In contrast, the UTI treatment alleviated tissue damage and inflammatory infiltration, suggesting notably better histological scores (Figure 2F,G). Intestinal goblet cells serve as an essential barrier, maintaining the integrity of mucosa and preventing pathogens from invading the mucosa to cause intestinal inflammation [29]. The AB/PAS staining of colonic sections demonstrated that UTI relieved the loss of goblet cells induced by DSS (Figure 2H,I). Briefly, UTI attenuated the pathological features in colitis mice and showed a similar effect as the 5-ASA treatment.

### 3.2. UTI Reduced Oxidative Stress and Modulated Colonic Inflammation in DSS-Induced Colitis

Inflammatory diseases are characterized by increased oxidative stress and overproduced inflammatory factors. On the one hand, we detected ROS signals in the mice by vivo imaging. As demonstrated, DSS induction invited higher-level ROS signal expression in the abdominal regions of the mice. At the same time, the UTI treatment markedly weakened the strength and range of the ROS signal (Figure 3A). On the other hand, we found a consistent inhibitory effect of UTI on the ROS through the measurement of ROS activity in the colon tissues (Figure 3B). In addition, UTI enhanced the expression of SOD (antioxidase) and reduced the levels of MDA (an indicator of peroxidation status) and MPO (a marker of neutrophil infiltration), indicating an ability to cause a shift towards the antioxidative environment (Figure 3C–E). In addition, RT-qPCR and the ELISA assay demonstrated that the pro-inflammatory cytokines of TNF-α, IL-1β, and IL-6 increased in number in the colitis tissues and sera, whereas UTI could reverse this pro-inflammatory phenotype (Figure 3F,G). Immunohistochemistry staining was carried out to visualize the accumulation of cytokines and cyclo-oxygenase-2 (COX-2) in situ in the DSS-attacked colonic mucous, suggesting that UTI produces therapeutic effects by modulating the presence of inflammatory mediators (Figure 3H). Immunofluorescence staining illustrated that UTI inhibited the infiltration of immune cells (CD4+ and CD45+) in the colon mucous contrast with the colitis group (Figure 3I). The above results showed the positive ability of UTI to regulate the inflammatory mediators in colitis.

### 3.3. UTI Maintained Intestinal Barrier Function and Integrity in DSS-Induced Colitis

The intestinal permeability of the mice was measured by the intragastric administration of FD4 after DSS induction for 7 days. All the DSS-induced mice displayed a higher fluorescence intensity in the sera than the healthy mice due to the disrupted gut epithelial barrier and increased intestinal permeability in colitis. The UTI treatment significantly diminished the fluorescence intensity in the sera, preventing the increase in intestinal permeability (Figure 4A). Intercellular tight junction (TJ) structures serve as an essential component of the gastrointestinal epithelium, providing a selectively permeable barrier by limiting pathogens and absorbing nutrients [30]. The disruption of the intestinal barrier is the primary pathological process of IBD. TJ proteins, including zonula occludens-1 (ZO-1), occludin, and claudins, safeguard the gut epithelial barrier integrity and excursion function. The expression of TJ proteins in the colon was evaluated by RT-qPCR and Western blotting (WB). Specifically, DSS-induced colitis markedly decreased the expression of ZO-1, occludin, and claudin-1 and dramatically increased the expression of claudin-2 at the mRNA levels; nevertheless, the abnormal expression was corrected by the UTI treatment (Figure 4B). The same results were obtained in the protein expression by quantification via WB (Figure 4C,D). Similarly, immunofluorescence staining demonstrated that UTI administration maintained the in situ expression of ZO-1 and occludin (Figure 4E). Given these outcomes, the UTI treatment corrected the abnormalities of the TJ proteins in colitis, and hence, preserved the intestinal barrier function.

### 3.4. UTI-Mediated Gut Barrier Protection Is Associated with JAK2/STAT3/SOCS1 Signaling Pathway

The transcriptome sequencing of colonic tissues from the UTI-treated and untreated colitis mice was employed to explore the specific molecular mechanisms that drive the anti-inflammatory effect of UTI. The Principle component analysis (PCA) of transcriptional profiling revealed the distinct clustering of the colitis group and the UTI group (Figure 5A). Based on the screening criteria of differentially expressed genes (DEGs) (log2 fold change > 1 and false discovery rate < 0.05), the UTI treatment significantly up-regulated 424 genes and downregulated 920 genes in the colon tissues (Figure 5B). Kyoto Encyclopedia of Genes and Genomes (KEGG) pathway analysis listed the top 15 most significantly different pathways, containing signaling pathways (Figure 5C). Given the negative WB results of crucial proteins in the PI3K-AKT signaling pathway and a weak association with inflammation of the cGMP-PKT signaling pathway, the JAK-STAT signaling pathway was used in subsequent validation analysis (Appendix A). There also has been some research reporting that UTI participates in the regulation of the JAK–STAT pathway to prevent organ injury in an animal inflammatory model [31]. Based on the sequencing results and previous reports, it is reasonable to assume that JAK-STAT is the master signaling pathway for UTI to attenuate colitis and avoid intestinal barrier impairment. Thus, we further analyzed the variation in gene expression in the JAK-STAT signaling pathway under UTI intervention to explore the exact mechanism. A total of 17 genes are differentially expressed, which are most related to inflammation and apoptosis (Figure 5D). Search tool for recurring instances of neighboring gene (STRING) analysis was employed to illustrate the interaction networks of the proteins translated from these genes (Figure 5E). A high confidence level (0.700) was set as the minimum required interaction score, and more nodes were added. The protein interaction networks revealed a strong interaction between SOCS1 and JAK2. SOCS (suppressor of cytokine signaling) family members are cytokine-inducible negative regulators of cytokine signaling and have an inhibitory effect on JAK and STAT. On the one hand, the RT-qPCR of colonic tissues was executed to verify the transcript levels of the filtered genes, and SOCS1 was found to be elevated to the greatest degree of the UTI group compared with that of the colitis group (Figure 5F). On the other hand, WB was performed to explore the changes in the JAK/STAT/SOCS pathway. As a result, the total JAK2 and STAT3 amounts did not differ significantly among the three groups. Compared with the control group, colitis dramatically upregulated the expression of phospho-JAK2 and phospho-STAT3 in the colon tissues, while the UTI treatment markedly inhibited the phosphorylation of JAK2 and STAT3 (Figure 5G). Statistical graphs delineated that the ratios of phospho-JAK2/JAK and phospho-STAT3/STAT3 were lower in the UTI group than those in the colitis group (Figure 5H). JAK2 and STAT3 constituted a membrane-to-nucleus signaling module, the phosphorylation of which participates in the expression and biological effect of various inflammatory mediators in colitis. Additionally, it is worth noting that the expression of SOCS1 increased significantly in the UTI group. Given the phosphorylation inhibition function of SOCS1, we speculated that UTI regulates the JAK2/STAT3/SOCS1 axis to reduce an inflammatory response and barrier impairment in colitis.

### 3.5. UTI Alleviates Gut Microbiota Dysbiosis in DSS-Induced Colitis Mice

Since gut microbial ecosystem disorder is closely related to the pathogenesis of UC, next, we performed the 16S rDNA gene high-throughput sequencing analysis of bacteria from the colon contents to further determine the moderating effects of UTI on the gut microbiota. The α-diversity analysis comprising Chao and Shannon indexes revealed that UTI significantly prevented the decrease in bacterial richness and diversity triggered by DSS induction (Figure 6A,B). For β-diversity, PCoA was applied based to the weighted Bray–Curtis distance, showing that the colitis group exhibited a shifted clustering of the gut microbiota structure that was significantly distinguished from the control group (*R*^2^ = 0.55, *p* < 0.001, Figure 6C); also, a partial overlap between the UTI group and the control and colitis groups was observed, respectively. These indicated that the gut microbiota structure in mice with colitis was significantly impacted by UTI. Likewise, a bipartite association network reflected the OTU variation between the different treatments (Figure 6D). One thousand and ninety-one OTUs were acquired from three groups, of which the control group overlapped three hundred and twenty-two OTUs with the UTI group and one hundred and sixty-one OTUs with the colitis group. Meanwhile, only 43 OTUs were shared by UTI and colitis. These results indicated that the microbiota composition in UTI lies somewhere between the control and colitis groups, but closer to the control rather than colitis group. The linear discriminant analysis (LDA) effect size (LEfSe) (LDA score > 4) demonstrated the fecal bacterial taxa with a significant difference among the groups (Figure 6E,F). At the genus level, *eubacterium_xylanophilum*, *Lachnospiraceae_NK4A136_group*, *Ileibacterium*, *Dubosiella*, and *Ligilactobacillus* dominated in the control group; *Turicibacter*, *Streptococcus*, *Bacteroides*, and *Escherichia_Shigella* dominated in the colitis group, while *Lactobacillus* and *unclassified_Lachnospiraceae* dominated in the UTI group.

Furthermore, we visualized the flora composition and compared the relative abundance of intestinal microbiota at the family and genus levels, respectively, (Figure 7A,C). At the family level, the relative abundance of *Bacteroidaceae*, *Deferribacteraceae*, *Enterobacteriaceae*, *Enterococcaceae*, and *Helicobacteraceae* increased, but that of *Eggerthellaceae*, *Lactobacillaceae*, *Muribaculaceae*, *Peptococcaceae*, and *Saccharimonadaceae* decreased in the colitis group (Figure 7B). At the genus level, the relative abundance of *Bacteroides*, *Enterococcus*, *Escherichia_Shigella*, *Helicobacter*, and *Mucispirillum* increased, but that of *Candidatus_Saccharimonas*, *Coriobacteriaceae_UCG_002*, *Lachnospiraceae_NK4A136_group*, and *Lactobacillus* in the colitis group decreased (Figure 7D). Additionally, there was a correlation between the abundance of intestinal flora above and inflammatory mediators (Appendix A). Summarized above, the UTI treatment modified colitis-induced changes in the strains and drove microbial ecology towards normality.

### 3.6. UTI Improved Microbial BA and SCFA Dysmetabolism in DSS-Induced Colitis

The recent studies have demonstrated that the metabolites interact with hosts, and microorganisms are involved in physiological and pathological processes [32]. To determine whether UTI could prevent DSS-induced gut metabolic disorders, we detected the BAs and SCFAs of the intestinal contents. The OPLS-DA of the detected metabolites in the three groups displayed samples in each group that were representative and comparable (Figure 8A). The metabolite clusters of the control group and colitis group lacked intersection, while the UTI group overlapped with the control group and the colitis group. All the examined metabolite levels are shown in a heat map (Figure 8B). Overall, the production level of metabolites in the colitis mice was significantly lower than that in the healthy mice, while the UTI treatment promoted metabolite production. On the one hand, UTI restored the content of BAs, including dehydrocholic acid (DHCA), deoxycholic acid (DCA), lithocholic acid (LCA), isolithocholic acid (ILCA), and dehydrolithocholic acid (DLCA) in the colitis mice (Figure 8C). On the other hand, the analysis of SCFAs demonstrated that UTI reversed the reduction of acetic acid (AA), isobutyric acid (IBA), butyric acid (BA), isovaleric acid (IVA), and valeric acid (VA) in the colitis mice (Figure 8D). Spearman’s correlation analysis between the metabolites and gut microbiota genera was employed. As is exhibited, the relative abundance of *Coriobacteriaceae_UCG_002*, *Lachnospiraceae_NK4A136_group*, and *Lactobacillus* was positively correlated with differential metabolites, including DHCA, DCA, LCA, AA, and BA (Figure 8E). Conversely, the relative abundance of *Bacteroides*, *Enterococcus*, *Escherichia_Shigella*, *Helicobacter*, and *Mucispirillum* was negatively correlated with these metabolites. Altogether, the UTI treatment was demonstrated to improve gut dysmetabolism through ameliorating gut microbiota dysbiosis in the DSS-induced colitis mice.

## 4. Discussion

UC endorsed as a “barrier organ disease” has attracted widespread attention due to its high associated risk for colon cancer [33]. In the therapeutical strategy for treating UC, how to diminish inflammation, while preserving barrier integrity and gut homeostasis, are the key points. So far, anti-inflammatory and immunosuppressive drugs are the main clinical agents that have been approved for UC treatment [34]. However, these drugs, such as 5-ASA or glucocorticoids, are associated with a high recurrence rate in patients with UC. Thereby, extensive research is warranted to develop novel and effective drugs for the precision and personalized treatment of UC.

UTI isolated from fresh urine is a urinary glycoprotein and protease inhibitor that suppresses a variety of serine proteases, including trypsin, plasmin, neutrophil elastase, and chymotrypsin. It is widely used to cure acute pancreatitis due to protease inhibition. UTI promotes lysosomal membrane stability and has been applied for treating acute circulatory failure (hemorrhagic shock, septic shock, and so forth) and in surgical operations on patients [35]. Meanwhile, UTI has also been shown to alleviate endothelial damage and inflammatory conditions of multiple organs in animal models [35,36,37,38]. In experimental models of craniocerebral, myocardial, renal, liver, and lung injury, UTI exerts potent antioxidant and anti-inflammatory effects [39,40,41]. In this study, we found that the treatment of UTI significantly alleviated the inflammatory manifestations in mice with colitis through inhibiting the JAK-STAT pathway. Meanwhile, UTI reduced intestinal barrier damage by maintaining the homeostasis of the gut microbiota. These findings indicate a novel clinical use of UTI in UC and potentially in other types of IBD, which also involve gut barrier defects and dysbiosis.

In colitis induced by DSS, the inflammatory mediators cooperatively promote inflammation and injury. The excessive production of ROS can induce cell degeneration and death and compromise the permeability of biomembranes by amplifying inflammatory signals, modifying lipoproteins to become pro-inflammatory, and causing DNA damage [42]. Additionally, the final products of lipid peroxidation initiated by ROS, like MDA, damage the TJ proteins through a series of reactions [43]. In the colitis mice, we found that UTI reduced the ROS activity, reduced the MDA yield, and promoted SOD generation, suggesting an antioxidant capacity. The accumulation of pro-inflammatory cytokines precludes the resolution of inflammation and results in colon destruction [44]. Our data demonstrated that UTI reduced the number of key pro-inflammatory cytokines in IBD, especially IL-1β and TNF-α. The IL-1 family is considered a vital initiator in colon inflammation due to the pathological activation of the innate immune system [45]. TNF-α deactivates the TJs and induces the apoptosis of epithelial cells, thus negatively affecting the barrier function and ulcer repair [46]. In its anti-inflammatory process, UTI displayed immune modulatory effects. The downregulation of activated immune cells in the intestinal epithelium was observed under the UTI treatment, reducing the release of pro-inflammatory cytokines (Figure 3I).

Through RNA sequencing and subsequent validation with molecular experiments, the anti-inflammatory action of UTI was demonstrated to be mediated by SOCS1, which inhibited the JAK2/STAT3 pathways, therefore lessening the signaling of cytokines. JAK, which is widely distributed in various tissues and cells, is a class of non-receptor protein tyrosine kinases, which works as the cytoplasmic signaling components of cytokine receptors [16]. JAK2 is activated through cytokine-mediated phosphorylation and downstream contributes to the phosphorylation of signal transducer and STAT proteins [47]. STAT3 is activated through phosphorylation in response to various cytokines, and thus plays a crucial role in many cellular processes, such as cell growth and apoptosis [48]. The JAK/STAT pathway serves as an essential molecular mechanism for specific cytokines, growth factors, and hormones to deliver bio-information into the nuclei of cells, and its dysregulation is associated with a variety of autoimmune diseases [49]. SOCS1 negatively regulates the JAK2/STAT3 pathway. Therefore, SOCS1 has crucial implications for maintaining immune homeostasis, especially under pathological conditions like autoimmunity [50]. The adjustment is accomplished in two ways: credit to an SH2 domain, where the SOCS proteins facilitate the degeneration of tyrosine-phosphorylated signaling intermediate, including phospho-JAK, phospho-STAT, and other phosphorylated receptors; SOCS1 can bind to and inactivate JAK2, thus preventing JAK2 from the phosphorylation of STAT3 [51]. Our data present compelling evidence that the UTI treatment promotes the expression of SOCS1, and thus inhibits the phosphorylation of JAK2 and STAT3 (Figure 5G,H). The regulatory effect of UTI on this pathway has been demonstrated in lung injury models, and we further identified the therapeutic role of this mechanism in colitis [31,52].

In addition, UTI protected gut homeostasis (Figure 6). An interrelation between IBD and enteric microorganisms has been shown through the convergence of observations and experiments [8,53]. Gut microbial dysbiosis in patients with colitis and animal models is mainly associated with *Lactobacillus*, *Bacteroides*, *Enterococcus*, *Streptococcus*, *Escherichia_Shigella*, *Mucispirillum*, etc. [54,55,56]. Notably, UTI improved the abundance of *Lactobacillus*, *Coriobacteriaceae_UCG_002*, and *Lachnospiraceae_NK4A136_group* in colitis mice. *Lactobacillus* is widely acknowledged as one of the most important intestinal probiotics, the supplementation of which has been shown to relieve symptoms of colitis and is increasingly comprehensively utilized [57,58]. *Lactobacillus* contributes to maintaining intestinal barrier function by increasing mucin production, immunomodulation, and pathogen inhibition [59]. The recovery of other strains benefited the switch of microbiome constitution from colitis to health. Regarding the intestinal flora downregulated by UTI, *Bacteroides* have been strongly correlated with some colitis-related genes [60]. *Enterococcus* and *Escherichia_Shigella* are considered pro-inflammatory [61]. Hence, the regulation of the gut microbiome is a method for UTI to exert anti-inflammatory and barrier-protective effects. However, whether this regulatory effect comes from the direct regulation of UTI or indirect regulation through alterations in the inflammatory environment still needs further investigation.

It has been identified that gut metabolites function as crucial molecular mediators between the microbiota and host [62,63]. The primary BAs undergo chemical modifications by gut bacteria to become secondary BAs [64]. The BAs work both as detergents, promoting nutrition assimilation, and as hormones, modulating nutrition metabolism [65]. BA signaling is deduced as a vital regulatory molecule of inflammation and immunity in the gut, which probably functions in the pathogenesis and treatment of colitis [66]. SCFAs metabolized by intestinal flora from indigestible sugars, such as dietary fiber, resistant starch, and oligosaccharides, have been confirmed to function therapeutically in animal models of colitis [67]. Also, several reports saw the curative effects of SCFA on patients with colitis [68]. UTI promoted the generation of metabolites, up-regulating the DCA and LCA of secondary Bas, as well as the BA and PA of SCFAs. There was compelling evidence with mouse models that DCA and LCA shaped the immune responses by regulating intestinal immune cells, such as macrophages, T regulatory cells, and effector T cells, which was achieved by activating the farnesoid X receptor and G-protein-coupled bile acid receptor pathway [69]. BAs produced from anaerobe, especially *Lactobacillus*, have been proven to modulate IL-10 receptor alpha subunit to inhibit TJ protein claudin-2 that increases barrier permeability [70,71]. Accordingly, we detected that the expression level of claudin-2 in the UTI group was depressed. Joint analysis demonstrated that the production of metabolites correlated with the microbiota regulated by UTI. Hence, we speculated that the metabolite is an essential interactive mediator between the microbiome and colon inflammation under the UTI treatment.

Based on our data, we propose an underlying mechanism for action of UTI in treating colitis (Figure 9). On one hand, UTI attenuated inflammation and reduced the severity of epithelial damage. The UTI treatment protected the TJ proteins and goblet cells from injury by attenuating both oxidative stress and the inflammatory response, suggesting a positive effect of UTI on mucosal barrier function. The mitigation of inflammatory processes was achieved by modulating the JAK2/STAT3/SOCS1 pathway to suppress cytokine signaling. On the other, UTI maintained gut homeostasis. UTI might restore the abundance of probiotic bacteria, especially *Lactobacillus*, and regulate the intestinal metabolites in colitis mice.

Although the therapeutic role of UTI in DSS-induced colitis was confirmed, there are still some limitations in this study. First, only one dosage of UTI was investigated in this study. The UTI dosage in previous murine experiments ranged from 5000 U/kg/d to 50,000 U/kg/d [28,39,40,41]. In this study, the colitis mice were treated with a lower dose (10,000 U/kg/d) and observed significant therapeutic effects. Therefore, our results are reliable, and follow-up studies are guaranteed to clarify the most appropriate dose for UC treatment. A further limitation of our study is that our conclusions are based mainly on one animal model. Although UTI’s anti-inflammatory and antioxidant effects have been confirmed, whether the therapeutic effect is also partly attributable to reducing enterotoxicity of DSS needs to be considered. DSS-induced colitis is a commonly used colitis model that leads to a pro-inflammatory phenotype with parallels with human UC. 2,4,6-trinitro-benzene sulfonic acid (TNBS) is another common chemical cause of IBD, and some of the immunological and histopathological characteristics resemble features of Crohn’s disease (another major phenotype of IBD) [72]. In addition to chemical induction, genetically engineered mice can be used to build colitis models. Given the heterogeneity of colitis, follow-up experiments should be carried out using multiple colitis models to evaluate the efficacy of UTI. Similarly, other mouse strains, such as Balb/c, should be included. Last, though UTI displayed a positive therapeutic effect in the mouse models of UC, there are still several impediments to the application of UTI in treatment and clinical trials. Intravenous injection is a usual route of UTI administration in a clinical setting, which is suitable for systematic rather than colonic inflammation. Optimal formulations or administration routines are necessary for the clinical application of UTI.

## 5. Conclusions

In summary, our data support the therapeutic role of UTI in the treatment of UC. UTI prominently ameliorated the symptoms of DSS-induced colitis in mice, with a blunted inflammatory response and the improved integrity of the intestinal barrier. One part of its protective effects is through maintaining the intestinal microflora balance, correcting the abnormalities of gut metabolites, and the other part is by modulating the JAK2/STAT3/SOCS1 signal pathway, and subsequently downregulating the inflammatory mediators.

## Figures and Tables

**Figure 1 antioxidants-13-00214-f001:**
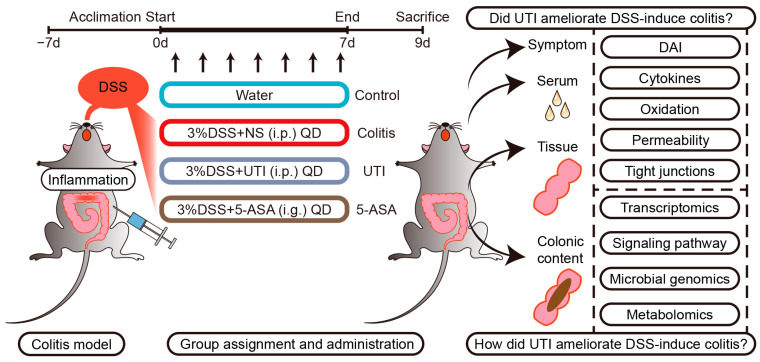
Schematic diagram of colitis model, drug administration, and tissue sampling.

**Figure 2 antioxidants-13-00214-f002:**
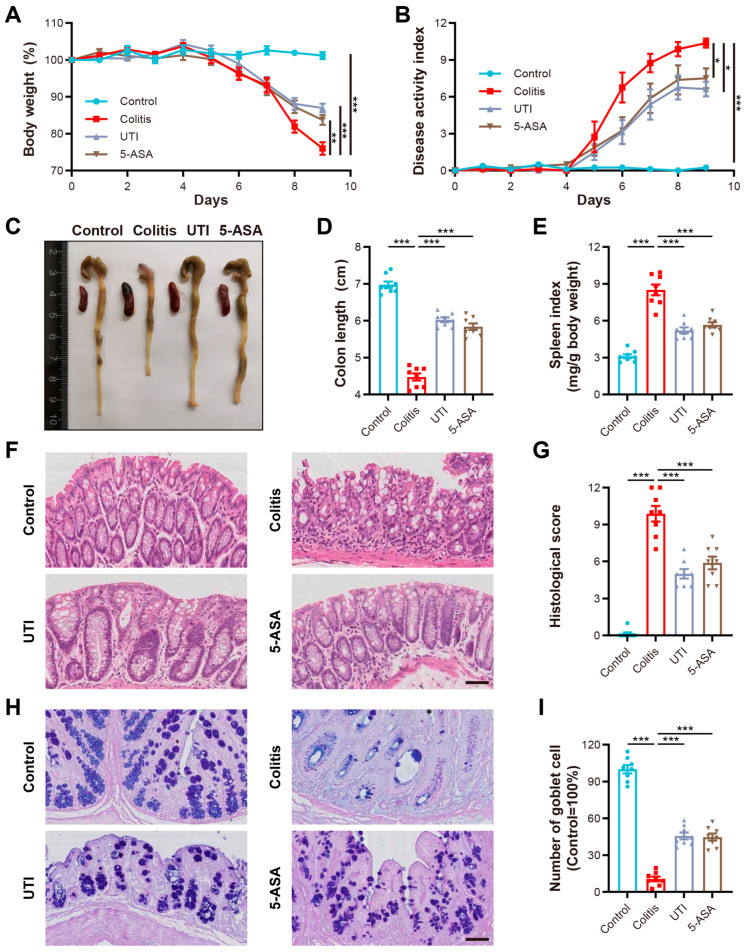
UTI treatment alleviates pathological symptoms in DSS-induced colitis. (**A**) Daily changes in body weight in each group. (**B**) Daily changes in DAI in each group. (**C**) Representative images of colon and spleen in each group. (**D**) Colon length in each group. (**E**) Spleen index (ratio of spleen weight and body weight) in each group. (**F**,**G**) Representative images of H&E-stained colon section and histological score in each group. (**H**,**I**) Representative images of AB/PAS-stained colon section and the number of goblet cells in each group. Scale bar = 100 µm. (*n* = 8). Data are presented as the mean ± SEM. * *p* < 0.05; ** *p* < 0.01; *** *p* < 0.001.

**Figure 3 antioxidants-13-00214-f003:**
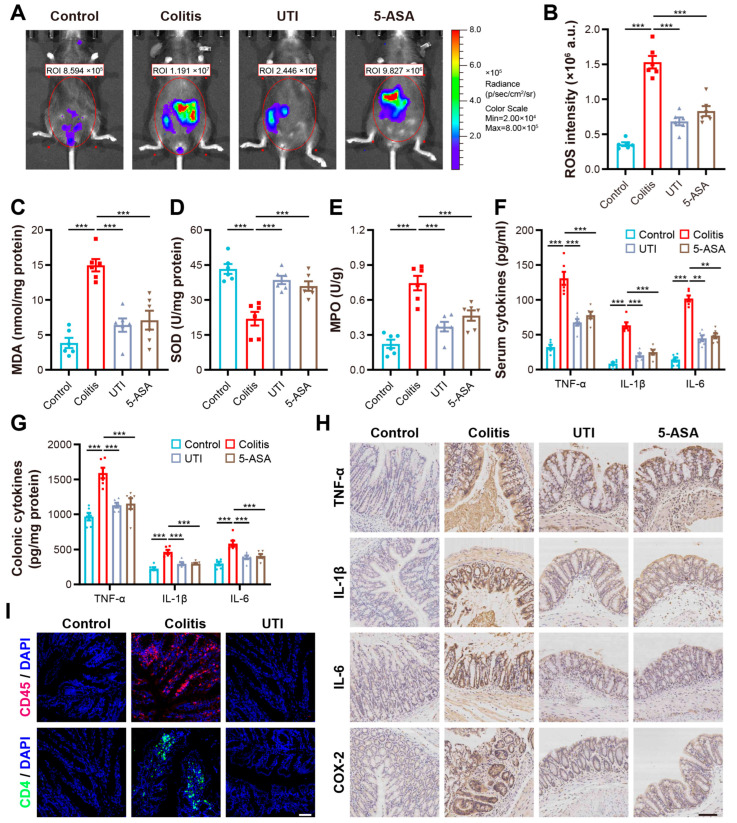
UTI treatment modulates oxidative stress and inflammation mediators in DSS-induced colitis. (**A**) In vivo imaging of abdominal ROS L-012 bioluminescent signals using an IVIS Spectrum CT system. (**B**) ROS activity in colonic tissues. (**C**) MDA activity in colonic tissues. (**D**) MPO activity in colonic tissues. (**E**) SOD activity in colonic tissues. (**F**) Relative mRNA expression of cytokines in colon tissues. (**G**) Cytokine (TNF-α, IL-1β, and IL-6) levels in sera from each group. (**H**) Immunohistochemical staining of inflammatory mediators (TNF-α, IL-1β, IL-6, and COX-2). Scale bar = 100 µm. (**I**) Immunofluorescent staining of CD4+, CD45+ cells infiltration in colon. Scale bar = 25 µm. (*n* = 6). Data are presented as the mean ± SEM. ** *p* < 0.01; *** *p* < 0.001.

**Figure 4 antioxidants-13-00214-f004:**
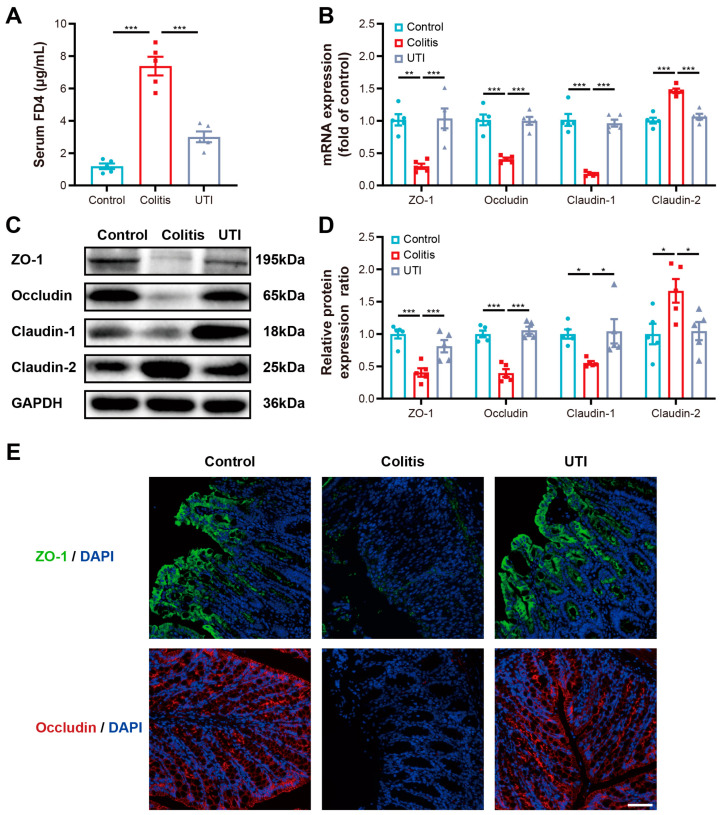
UTI restores gut barrier function in DSS-induced colitis. (**A**) Serum concentrations of fluorescein isothiocyanate-dextran. (**B**) Relative mRNA expression of tight junction proteins (ZO-1, Occludin, Claudin-1, and Claudin-2) in colon tissues. (**C**,**D**) Representative Western blotting images and the relative expression levels of tight junction proteins (ZO-1, Occludin, Claudin-1, and Claudin2) in colon tissues. (**E**) Immunofluorescence images showing in situ expression of tight junction proteins (ZO-1 and Occludin). Scale bar = 25 µm. (*n* = 5). Data are presented as the mean ± SEM. * *p* < 0.05; ** *p* < 0.01; *** *p* < 0.001.

**Figure 5 antioxidants-13-00214-f005:**
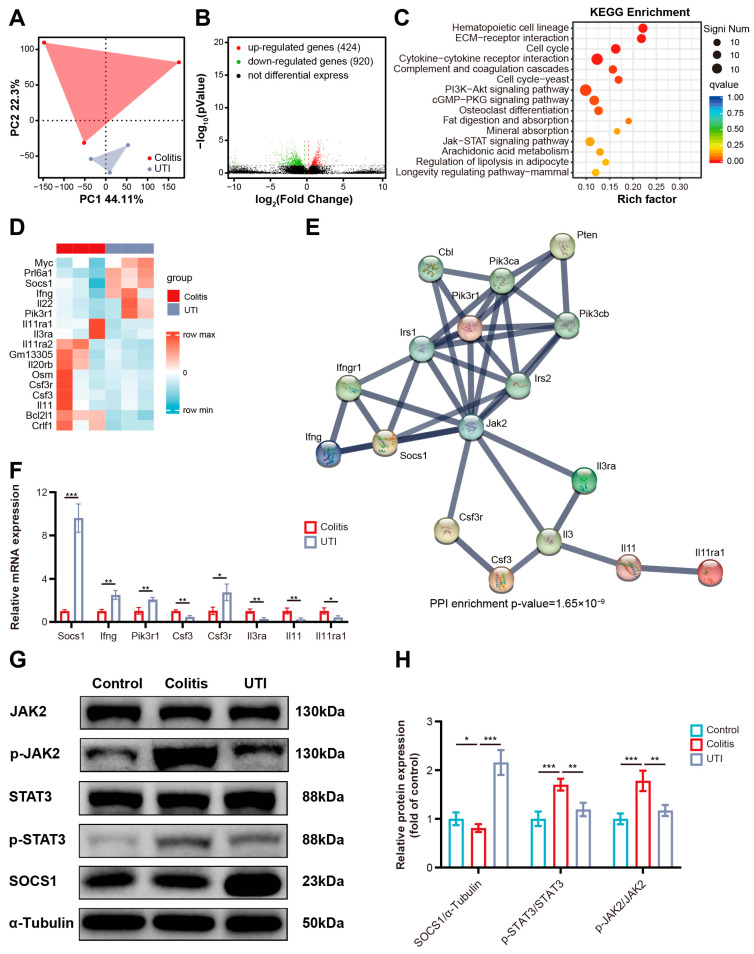
UTI interferes with JAK2/STAT3/SOCS1 pathways in DSS-induced colitis. (**A**) Principal component analysis of transcriptional profiling in colon tissues. (**B**) Volcano plot of differentially expressed genes. (**C**) KEGG enrichment analysis of the top 15 significantly changed pathways. (**D**) Heat map of differentially expressed genes in the JAK-STAT signaling pathway. (**E**) Search tool for recurring instances of neighboring genes network visualization of the genes in differentially expressed genes in the JAK-STAT signaling pathway. Edges represent protein–protein associations. (**F**) RT-qPCR assay for the key differentially expressed genes in the JAK-STAT signaling pathway. (**G**,**H**) Representative Western blotting images and the relative expression levels of JAK2/STAT3/SOCS1 pathways proteins. (*n* = 3). Data are presented as the mean ± SEM. * *p* < 0.05; ** *p* < 0.01; *** *p* < 0.001.

**Figure 6 antioxidants-13-00214-f006:**
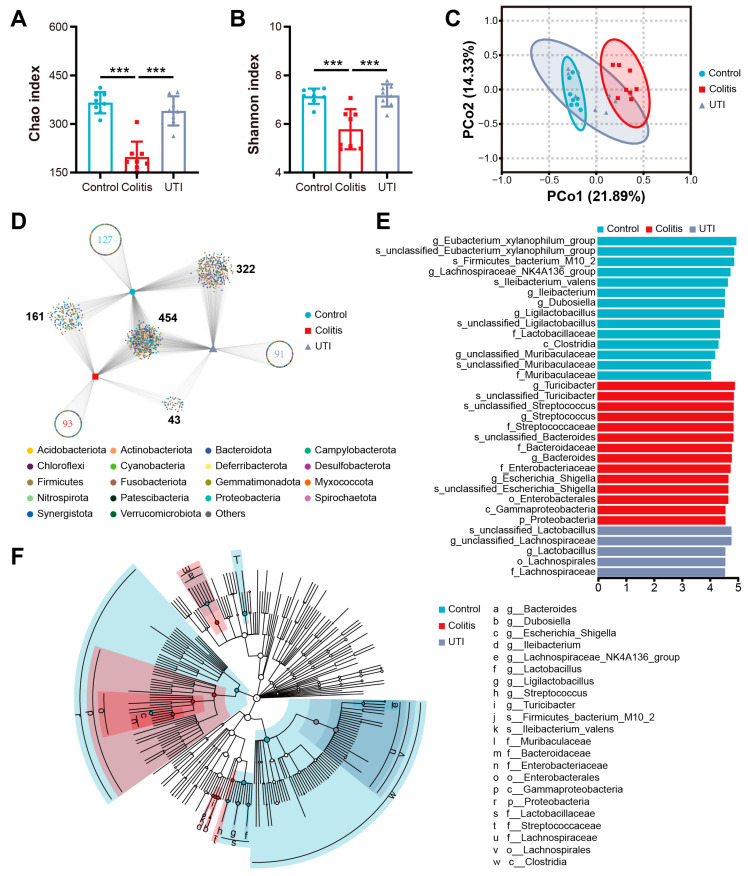
UTI relieves intestinal microbiota dysbiosis in DSS-induced colitis mice. (**A**,**B**) Measurement of alpha-diversity indexes: Chao and Shannon. (**C**) Principal coordinate analysis of the Bray–Curtis distance based on OTUs. (**D**) The bipartite association network represents overlapping and independent phylum of OTUs among different treatment groups. (**E**) LDA scores show significant bacterial differences among different treatment groups (log LDA > 4.0). (**F**) A cladogram using the LEfSe method shows the phylogenetic distribution of the colonic microbes found to be significantly associated with DSS administration and UTI treatment. (*n* = 8). Data are presented as the mean ± SEM.; *** *p* < 0.001.

**Figure 7 antioxidants-13-00214-f007:**
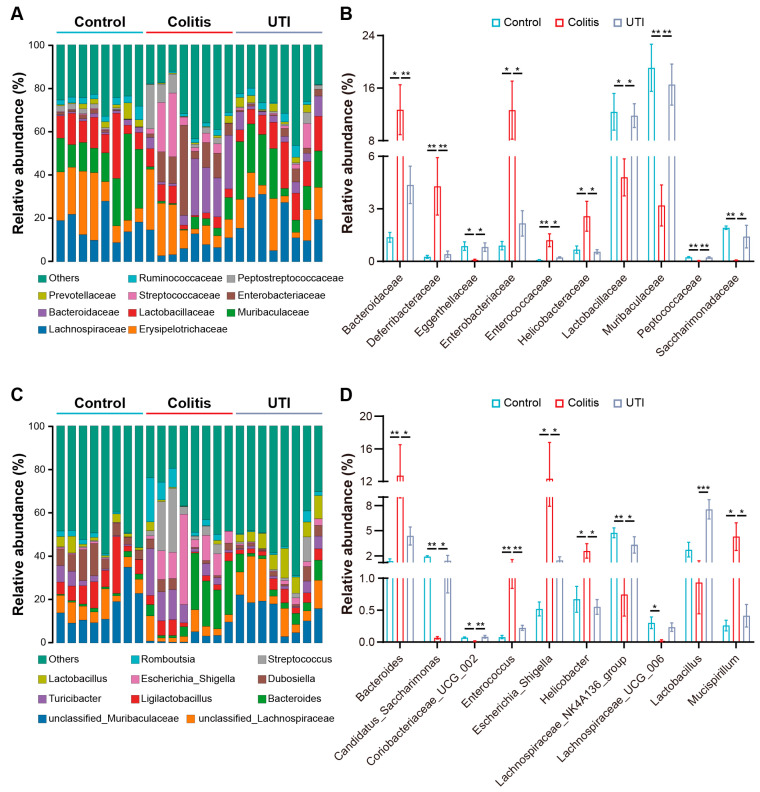
UTI modulates intestinal microbiota composition in DSS-induced colitis mice. (**A**) Composition of intestinal microbiota at the family level. (**B**) Relative abundance of the significantly altered bacteria at the family level in each group. (**C**) Composition of intestinal microbiota at the genus level in each group. (**D**) Relative abundance of the significantly altered bacteria at the genus level in each group. (*n* = 8). Data are presented as the mean ± SEM. * *p* < 0.05; ** *p* < 0.01; *** *p* < 0.001.

**Figure 8 antioxidants-13-00214-f008:**
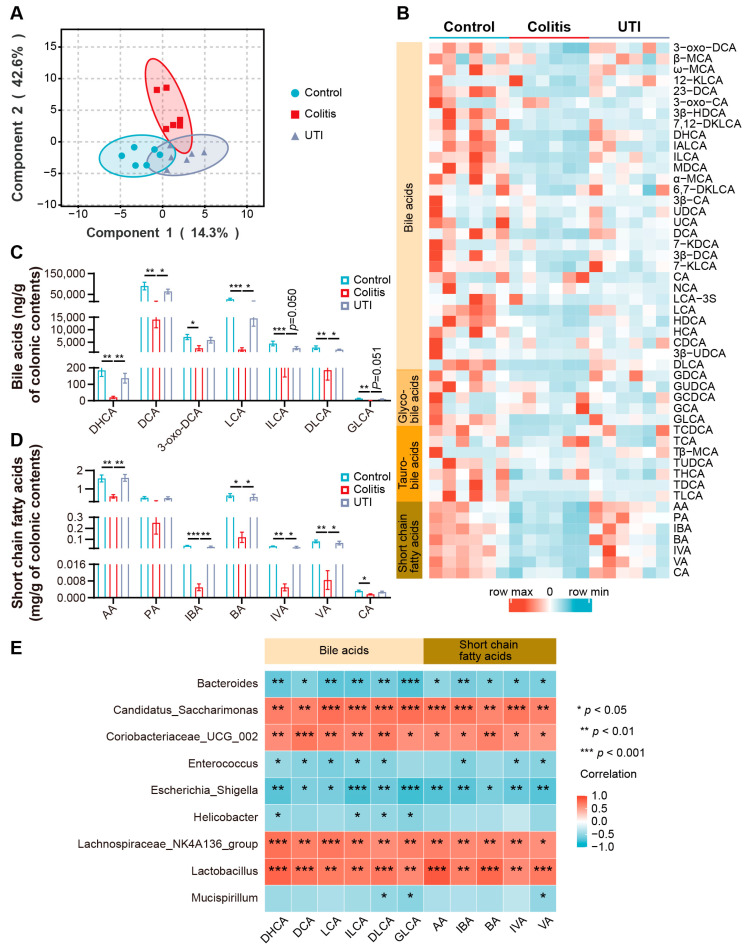
UTI regulates the expression of BAs and SCFAs in DSS-induced colitis mice. (**A**) Orthogonal partial least squares discriminant analysis of the metabolites. (**B**) Heatmap of the metabolites. (**C**) Comparison of the significantly changed BAs in each group. (**D**) Comparison of the significantly changed SCFAs in each group. (**E**) Heat map showing the correlation between gut metabolites and microbiota. (*n* = 6). Data are presented as the mean ± SEM. * *p* < 0.05; ** *p* < 0.01; *** *p* < 0.001.

**Figure 9 antioxidants-13-00214-f009:**
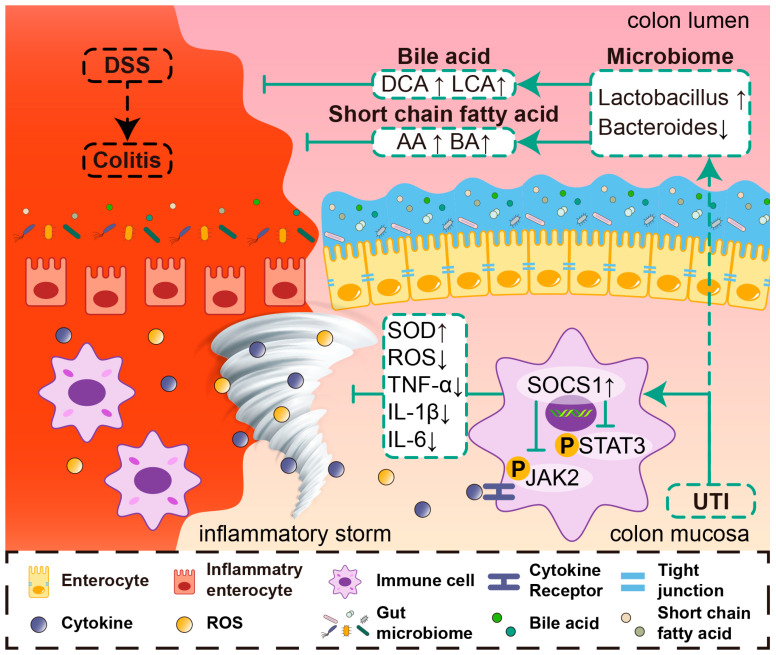
Illustration of the molecular mechanism of UTI in treating colitis in mice. UTI treatment regulates JAK2/STAT3/SOCS1 pathway, improves microbiota structure, and promotes metabolites generation. These mechanisms protect the intestinal barrier, reduce colonic inflammation, and maintain gut homeostasis.

## Data Availability

The original contributions presented in the study are included in the article and Appendix A, further inquiries can be directed to the corresponding authors. The raw sequencing data supporting the conclusions of this article will be made available by the authors on request.

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
