# Peer review of "Integrative Multiomics Profiling Unveils the Protective Function of Ulinastatin against Dextran Sulfate Sodium-Induced Colitis"

_antioxidants, 2024, doi:10.3390/antiox13020214_

Round 1
Reviewer 1 Report
Comments and Suggestions for Authors
Yu and colleagues studied the therapeutic role of ulinastatin (UTI), an anti-inflammatory bioagent, and its associated mechanisms in treating colitis. The authors employed qRT-PCR, metabolomics, 16S Illumina sequencing, and transcriptomics in this study. The data is sufficient and solid. I think it almost hits the publish standard.
However, there is one issue: What is function of 5-ASA? The authors need to introduce it in Introduction and M&M part. Without this part, it is hard to understand the design for the positive control in this study. Otherwise, all data is good!
The discussion needs some deeper discuss on the UC treatment and the improvement to the use of UTI. Also, the limitation of applying UTI in treatment and clinical trail.
Author Response
Dear Reviewer,
Thank you very much for taking the time to review this manuscript. Please find the detailed responses below and the corresponding revisions/corrections highlighted/in track changes in the re-submitted files.
| 
 Questions for General Evaluation  | 
 Reviewer’s Evaluation  | 
 Response and Revisions  | 
| 
 Does the introduction provide sufficient background and include all relevant references?  | 
 Can be improved  | 
 We revised background, and corresponding response is included in the point-by-point response letter.  | 
| 
 Are all the cited references relevant to the research?  | 
 Yes  | 

  | 
| 
 Is the research design appropriate?  | 
 Yes  | 

  | 
| 
 Are the methods adequately described?  | 
 Can be improved  | 
 We modified the methods, and corresponding response is included in the point-by-point response letter.  | 
| 
 Are the results clearly presented?  | 
 Yes  | 

  | 
| 
 Are the conclusions supported by the results?  | 
 Yes  | 

  | 
Point-by-point response to Comments and Suggestions for Authors
Yu and colleagues studied the therapeutic role of ulinastatin (UTI), an anti-inflammatory bioagent, and its associated mechanisms in treating colitis. The authors employed qRT-PCR, metabolomics, 16S Illumina sequencing, and transcriptomics in this study. The data is sufficient and solid. I think it almost hits the publish standard.
Comment 1: However, there is one issue: What is function of 5-ASA? The authors need to introduce it in Introduction and M&M part. Without this part, it is hard to understand the design for the positive control in this study. Otherwise, all data is good!
Response 1: Thank you for pointing this out. 5-ASA was used as the control drug in our study, the function of which needs to be elaborated to improve the logicality and reliability of the results. 5-ASA, also known as mesalazine, has been used for over 40 years to treat inflammatory bowel disease (ulcerative colitis [UC] and mild to moderate Crohn's disease [CD]). 5-ASA belongs to the family of nonsteroidal anti-inflammatory drugs (NSAIDs) and is an anti-oxidant with high biological activity among NSAIDs. As the first-line anti-inflammatory therapy for UC, 5-ASA was used as a positive control to evaluate the efficacy of the experimental drug in most of the relevant literature. Therefore, we chose 5-ASA as the positive control drug. In UC treatment, 5-ASA functions through anti-oxidant and anti‑inflammatory activity, whose main mechanisms involve inhibiting lipid peroxidation, scavenging reactive oxygen species, increasing the expression of the anti-oxidant, and controlling cytokine secretion. The PI3K/Akt, Keap1/Nrf2, and NF-κB signaling pathways have been implicated in 5-ASA function (Beiranvand, Mohammad, Inflammopharmacology, 2021, DOI:10.1007/s10787-021-00856-1). Hence, we examined the role of UTI in these aspects in the manuscript and found that UTI had anti-oxidant and anti-inflammatory functions similar to 5-ASA. Meanwhile, UTI exerted therapeutic effects through different signaling pathways (maintaining intestinal microflora balance, correcting the abnormalities of gut metabolites, and modulating the JAK2/STAT3/SOCS1 signal pathway). Besides, UTI has been shown to have intestinal barrier restorative ability that 5-ASA does not have. These findings suggest the potential of UTI in the treatment of UC. To address this comment, we have included a detailed introduction to 5-ASA in the Introduction and M&M part of the revised manuscript.
Comment 2: The discussion needs some deeper discuss on the UC treatment and the improvement to the use of UTI. Also, the limitation of applying UTI in treatment and clinical trail.
Response 2: Although UTI showed a positive therapeutic effect in mouse models of UC, there are still several limitations and impediments to the application of UTI in UC treatment. First, UTI is currently used to treat sepsis and pancreatitis. Whether the current drug concentrations are appropriate for treating UC remains unknown. The appropriate concentration of UTI for UC treatment needs to be explored in more in-depth multi-dose studies. Secondly, intravenous injection is a usual route of UTI administration in a clinical setting. For colon inflammation treatment rather than systematic inflammation, further studies are needed to prepare more efficient and safer UTI formulations for local delivery. Besides, as a broad-spectrum protease inhibitor, long-term use of UTI may raise potential function disturbance in the body. Despite promising curative effects for mouse models, UTI has several clinical trial and application limitations. To address this comment, we have expanded the discussion to describe the improvement and limitations of UTI application in UC treatment.
Reviewer 2 Report
Comments and Suggestions for Authors
This study evaluated the effect of ulinastatin (UTI), an anti-inflammatory bioagent, and its associated mechanisms in treating colitis-induced Dextran sulfate sodium in mice. The authors evaluated that gut barrier defect and inflammatory manifestations of colitis were determined via histological and molecular experiments, and transcriptomics, metagenomics, and metabolomics were employed to explore possible mechanisms underlying the effects of UTI. In summary, this MS suggested that UTI was effective in treating colitis and supported a potential use of UTI treatment for patients with ulcerative colitis. This is an important contribution to the understanding of ulcerative colitis and gives new information. I have some comments, explained below. I hope my comments are beneficial for improving this research.
Comments
(1) Ulcerative colitis: The authors' logic is based on the assumption that UC is the symptom of DSS-induced colitis in the mice used in this experiment. However, do the mice really have ulcerative colitis? The authors should confirm from the pathological section images of the colon whether they are really causing ulcerative colitis symptoms.
(2) Section 2.5 and 2.7: On how many days after DSS administration was the in vivo imager and permeability tested? Figure 1 shows that these tests are marked as having been performed after autopsy. However, these tests cannot be tested unless the mice are alive.
(3) Section 2.8:Please indicate the antibodies used in the immunostaining.
(4) Section 2.12: Please indicate the database used when attributing sequence information; Qiime2 should use either greengene or silva.
(5) Section 2.14, Fig. 2B and 2E: The results in Fig. 2B and 2E are on an ordinal rather than an interval scale. Therefore, the Tukey method, a parametric test, is inappropriate. The authors need to change to the correct statistical method.
(6) Fig.9:Do UTIs really affect Lactobacillus and Bacteroides? Figure 9 shows arrows from UTI to these microorganisms. The gut microbiota data shows no bacteria changed only in the UTI-treated group; could the change in gut microbiota due to UTI be a result of UTI suppressing the exacerbation of colitis symptoms caused by DSS, and thus the gut microbiota did not change much? In this experiment, there is no group that received only UTI without DSS, so we do not know whether the bacterial flora was changed by UTI.
(7) DSS: How did UTI alleviate the toxicity of DSS, suggesting the involvement of Dectin 1 in the exacerbation of colitis symptoms caused by DSS (Cell Host Microbe. 2015 Aug 12;18(2):183-97. doi: 10.1016/j.chom.2015.07.003.)? The results of this study show that UTI mitigates many of the changes caused by DSS. However, we believe that this is a secondary change that occurred because of the mitigation of DSS toxicity. Therefore, I think a discussion on the mitigation of DSS toxicity by UTI is needed.
Author Response
Dear Reviewer,
Thank you very much for taking the time to review this manuscript. Please find the detailed responses below and the corresponding revisions/corrections highlighted/in track changes in the re-submitted files.
| 
 Questions for General Evaluation  | 
 Reviewer’s Evaluation  | 
 Response and Revisions  | 
| 
 Does the introduction provide sufficient background and include all relevant references?  | 
 Yes  | 

  | 
| 
 Are all the cited references relevant to the research?  | 
 Yes  | 

  | 
| 
 Is the research design appropriate?  | 
 Yes  | 

  | 
| 
 Are the methods adequately described?  | 
 Can be improved  | 
 We modified the methods, and corresponding response is included in the point-by-point response letter.  | 
| 
 Are the results clearly presented?  | 
 Can be improved  | 
 We revised the results, and corresponding response is included in the point-by-point response letter.  | 
| 
 Are the conclusions supported by the results?  | 
 Can be improved  | 
 We revised the results, and corresponding response is included in the point-by-point response letter.  | 
Point-by-point response to Comments and Suggestions for Authors
This study evaluated the effect of ulinastatin (UTI), an anti-inflammatory bioagent, and its associated mechanisms in treating colitis-induced Dextran sulfate sodium in mice. The authors evaluated that gut barrier defect and inflammatory manifestations of colitis were determined via histological and molecular experiments, and transcriptomics, metagenomics, and metabolomics were employed to explore possible mechanisms underlying the effects of UTI. In summary, this MS suggested that UTI was effective in treating colitis and supported a potential use of UTI treatment for patients with ulcerative colitis. This is an important contribution to the understanding of ulcerative colitis and gives new information. I have some comments, explained below. I hope my comments are beneficial for improving this research.
Comments 1: Ulcerative colitis: The authors' logic is based on the assumption that UC is the symptom of DSS-induced colitis in the mice used in this experiment. However, do the mice really have ulcerative colitis? The authors should confirm from the pathological section images of the colon whether they are really causing ulcerative colitis symptoms.
Response 1: We agree with this comment. The gold standard for ulcerative colitis is pathological diagnosis. We presented the representative pathological section images of the colon in Figure 2F. HE staining of histological sections showed that DSS treatment triggers mucosal inflammation and injuries. In the colitis group of pathological images, inflammatory cell infiltration, crypt damage, and surface epithelial destruction could be observed, consistent with the pathological features of ulcerative colitis. Additionally, AB/PAS staining of the pathological sections demonstrated that intestinal goblet cells decreased, indicating the destruction of the intestinal epithelial barrier under DSS treatment (Figure 2H). In the revised manuscript, we supplemented and improved the description of pathological sections. Meanwhile, we mentioned that pathological section images of the colon confirmed the successful establishment of the colitis model.
Comments 2: Section 2.5 and 2.7: On how many days after DSS administration was the in vivo imager and permeability tested? Figure 1 shows that these tests are marked as having been performed after autopsy. However, these tests cannot be tested unless the mice are alive.
Response 2: Thank you for pointing this out. In vivo imaging was conducted on day 7 after DSS administration. Experimental mice were gavaged with fluorescein isothiocyanate-dextran on day 9. 4 hours after gavage, mice were euthanized. Then, serum was collected for the permeability test, and tissues were collected for other investigation. In the revised manuscript, we have optimized the description of sections 2.5 and 2.7 to emphasize the accuracy of experimental methods.
Comments 3: Section 2.8: Please indicate the antibodies used in the immunostaining.
Response 3: We have reorganized the supplemental files and listed the antibodies used in the immunostaining in the Supplement Table S2.
Comments 4: Section 2.12: Please indicate the database used when attributing sequence information; Qiime2 should use either greengene or silva.
Response 4: Thank you for your comments which rounded out our manuscript. In section 2.12, operational taxonomic units were classified using the SILVA bacteria taxonomy. We have, accordingly, modified the description in section 2.12 to emphasize this point.
Comments 5: Section 2.14, Fig. 2B and 2E: The results in Fig. 2B and 2E are on an ordinal rather than an interval scale. Therefore, the Tukey method, a parametric test, is inappropriate. The authors need to change to the correct statistical method.
Response 5: Thank you for your comment. The results in Fig. 2B and 2E are on an ordinal scale. If the data did not fulfill the prerequisites of parametric statistics, a Kruskal–Wallis test followed by Dunn's multiple comparisons was performed. We have, accordingly, revised the description in section 2.14 to emphasize this point. We also rechecked the results using suitable statistical methods.
Comments 6: Fig.9:Do UTIs really affect Lactobacillus and Bacteroides? Figure 9 shows arrows from UTI to these microorganisms. The gut microbiota data shows no bacteria changed only in the UTI-treated group; could the change in gut microbiota due to UTI be a result of UTI suppressing the exacerbation of colitis symptoms caused by DSS, and thus the gut microbiota did not change much? In this experiment, there is no group that received only UTI without DSS, so we do not know whether the bacterial flora was changed by UTI.
Response 6: Thank you for your constructive comment. We agree. Based on Fig. 7, we found that the probiotic abundance of the UTI group was close to that of the control group but not higher than that of the control group. The absence of the group that received only UTI without DSS hindered us from attributing the changes in microbiota directly to the regulatory role of UTI. The microbiota changes induced by UTI administration may be attributed to its regulation of the inflammatory environment of the gut. In the future follow-up experiment, we plan to set the relevant control group to analyze the regulatory effect of UTI on the gut flora. In the present study, based on the gut microbiota data, we speculate that the regulation of intestinal microbiota is one of the UTI therapeutic mechanisms. Accordingly, we have modified Fig. 9 to make the conclusion more reasonable. To address the comment, we have added limitations to the microbiota results in the discussion section and modified the results.
Comments 7: DSS: How did UTI alleviate the toxicity of DSS, suggesting the involvement of Dectin 1 in the exacerbation of colitis symptoms caused by DSS (Cell Host Microbe. 2015 Aug 12;18(2):183-97. doi: 10.1016/j.chom.2015.07.003.)? The results of this study show that UTI mitigates many of the changes caused by DSS. However, we believe that this is a secondary change that occurred because of the mitigation of DSS toxicity. Therefore, I think a discussion on the mitigation of DSS toxicity by UTI is needed.
Response 7: Thanks for sharing the literature to give us a deeper understanding of the colitis model. Although DSS is an extensively used chemical inducer, the exact mechanism of its colitogenicity remains incompletely understood. It is generally believed that DSS exerts toxicity toward intestinal epithelial cells of the basal crypts and triggers colitis (Nature Protocols, Wirtz, Stefan, 2017, DOI: 10.1038/nprot.2017.044). Although we cannot completely deny that UTI mitigated the effect of DSS toxicity on experimental results, it is observed that ulcerative colitis symptoms were ameliorated under UTI treatment from Figure 2 to Figure 4. Meanwhile, UTI exhibited anti-oxidant and anti-inflammatory properties in multiple detections, which is essential in treating colitis. More model validation might resolve this confusion. Therefore, to address this comment, we added the influence of DSS toxicity on the study results in the discussion section.
Reviewer 3 Report
Comments and Suggestions for Authors
This interesting study is based on the previous work of Jiang LY et al (Ref #23 of this manuscript) showing the protective effects of an anti-inflammatory bioagent, ulinastatin (UTI) on intestinal mucosal barrier function in rats with sepsis. The authors, therefore, aimed at delineating the mechanistic pathways underlying the therapeutical effect of UTI in attenuating colitis in mice induced by dextran sulfate sodium (DSS). The results showed that UTI ameliorated the symptoms of DSS-induced colitis in mice by alleviating the increased levels of pro-inflammatory cytokines and improving the integrity of the intestinal barrier. Moreover, UTI-mediated attenuation of increased pro-inflammatory cytokines levels was found to occur via JAK2/STAT3/SOCS1 signal pathway. UTI was shown to upregulate the expression of SOCS1, which subsequently inhibited phosphorylation of JAK2 and STAT3. 16S rRNA sequencing revealed that UTI maintained a more stable intestinal flora, protecting the gut from dysbiosis in colitis. UTI increased the abundance of beneficial Lactobacillus, while reducing the pathogenic bacteria. Furthermore, metabolomics analysis showed that UTI facilitated the production of certain BAs (secondary bile acids, DCA, LCA) and SCFAs (acetic acid (AA), butyric acid (BA)), which are essential for intestinal homeostasis. These findings provide evidence for ulinastatin (UTI) as a potential therapeutic drug in the treatment of colitis/IBD by modulating JAK2/STAT3/SOCS1 signaling, and subsequently downregulating inflammatory mediators as well as maintaining intestinal microflora balance and correcting the abnormalities of gut metabolites. Overall, the manuscript is straight-forward, easily understandable and the results are clearly presented. However, the authors should consider the following suggestions:
1. It is a general consensus that JAK/STAT pathway is linked to the pro-inflammatory cytokine IFNg signaling. Did the authors assess the levels of IFNg in the serum and colon of DSS colitis mice? Does IFNg also play a role in mediating the inflammatory effects?
2. Linear discriminant analysis (LDA) effect size (LEfSe) (LDA score>4) demonstrated the fecal bacterial taxa with a significant difference among groups. For example, at the genus level, Turicibacter was shown to be dominant in the colitis gp. However, it is somewhat surprising that there was no mention of Turicibacter in Figure 7D.
3. Since Turicibacter is known to modifiy bile acids in the host via bile salt hydrolases (BSH) (Lynch JB, Nat Commun, 14, 3669, 2023), did the authors perform correlation studies between Turicibacter and bile acids?
4. Suppl. Table 2: Details of SOCS1 (used for western blotting), CD4+, CD45+ (used for IF staining) and IL1b, IL6, TNFa and COX2 (used for IHC staining) are missing
5. Figure legends 2 and 3- Please remove *P<0.05
6. Figure 9 (proposed model)- Replace colon ‘cavity’ with ‘lumen’
7. It is ‘Feces’ not ‘Fece’
8. It is not clear whether Feces (fecal pellets) as mentioned in Figure 8C, D or colonic contents (as mentioned in Methods) were utilized to measure BAs and SCFAs
9. Please provide a rationale as to why only female C57BL/6J mice were used for this study?
Comments on the Quality of English LanguagePlease carefully proof-read the manuscript for typo and grammatical errors
Author Response
Dear Reviewer,
Thank you very much for taking the time to review this manuscript. Please find the detailed responses below and the corresponding revisions/corrections highlighted/in track changes in the re-submitted files.
| 
 Questions for General Evaluation  | 
 Reviewer’s Evaluation  | 
 Response and Revisions  | 
| 
 Does the introduction provide sufficient background and include all relevant references?  | 
 Yes  | 

  | 
| 
 Are all the cited references relevant to the research?  | 
 Yes  | 

  | 
| 
 Is the research design appropriate?  | 
 Yes  | 

  | 
| 
 Are the methods adequately described?  | 
 Can be improved  | 
 We modified the methods, and corresponding response is included in the point-by-point response letter.  | 
| 
 Are the results clearly presented?  | 
 Yes  | 

  | 
| 
 Are the conclusions supported by the results?  | 
 Yes  | 

  | 
| 
 Is it necessary to include study limitations in the discussion?  | 
 Can be improved  | 
 We modified the discussion, and corresponding response is included in the point-by-point response letter.  | 
Point-by-point response to Comments and Suggestions for Authors
This interesting study is based on the previous work of Jiang LY et al (Ref #23 of this manuscript) showing the protective effects of an anti-inflammatory bioagent, ulinastatin (UTI) on intestinal mucosal barrier function in rats with sepsis. The authors, therefore, aimed at delineating the mechanistic pathways underlying the therapeutical effect of UTI in attenuating colitis in mice induced by dextran sulfate sodium (DSS). The results showed that UTI ameliorated the symptoms of DSS-induced colitis in mice by alleviating the increased levels of pro-inflammatory cytokines and improving the integrity of the intestinal barrier. Moreover, UTI-mediated attenuation of increased pro-inflammatory cytokines levels was found to occur via JAK2/STAT3/SOCS1 signal pathway. UTI was shown to upregulate the expression of SOCS1, which subsequently inhibited phosphorylation of JAK2 and STAT3. 16S rRNA sequencing revealed that UTI maintained a more stable intestinal flora, protecting the gut from dysbiosis in colitis. UTI increased the abundance of beneficial Lactobacillus, while reducing the pathogenic bacteria. Furthermore, metabolomics analysis showed that UTI facilitated the production of certain BAs (secondary bile acids, DCA, LCA) and SCFAs (acetic acid (AA), butyric acid (BA)), which are essential for intestinal homeostasis. These findings provide evidence for ulinastatin (UTI) as a potential therapeutic drug in the treatment of colitis/IBD by modulating JAK2/STAT3/SOCS1 signaling, and subsequently downregulating inflammatory mediators as well as maintaining intestinal microflora balance and correcting the abnormalities of gut metabolites. Overall, the manuscript is straight-forward, easily understandable and the results are clearly presented.
Comments 1: It is a general consensus that JAK/STAT pathway is linked to the pro-inflammatory cytokine IFNg signaling. Did the authors assess the levels of IFNg in the serum and colon of DSS colitis mice? Does IFNg also play a role in mediating the inflammatory effects?
Response 1: Thank you for your constructive comments. Cytokines exert their biological effects by binding to specific receptors on the cell surface by the JAK/STAT pathway. The JAK2/STAT3 signaling pathway we detected in this manuscript mainly relates to the IL-6R and IFN-R families (Coskunn Mehmet, Pharmacological Research, 2013, DOI: 10.1016/j.phrs.2013.06.007). Generally, patients with Crohn's disease (CD) have increased Th1 cells and are characterized by having mucosal Th1/Th17-type responses, whereas patients with ulcerative colitis (UC) favor a mucosal Th2-mediated inflammation. Th1 differentiation is characterized by the expression of IFNg, whereas Th2 cells release cytokines such as IL-4, IL-5, and IL-13. Hence, IFNg may play a more prominent role in CD treatment. Since we constructed a DSS-induction model closer to UC, we chose other cytokines, such as TNF-α and IL-6, for serum and colon detection. Inspired by your comments, future study is recommended to investigate the levels of IFNg in follow-up experiments and explore the regulatory mechanisms of IFNg in colitis.
Comments 2: Linear discriminant analysis (LDA) effect size (LEfSe) (LDA score>4) demonstrated the fecal bacterial taxa with a significant difference among groups. For example, at the genus level, Turicibacter was shown to be dominant in the colitis gp. However, it is somewhat surprising that there was no mention of Turicibacter in Figure 7D.
Response 2: In fact, when it comes to statistical analysis, we found that the abundance of Turicibacter existed differences between groups by Tukey’s multiple comparisons test (control vs. colitis P=0.0496, UTI vs. colitis P=0.0282). Given the limited space of the picture, we demonstrated 10 bacteria that correlated with BAs and SCFAs in Figure 7D. The abundance data of Turicibacter is presented in the attachment.
Comments3: Since Turicibacter is known to modifiy bile acids in the host via bile salt hydrolases (BSH) (Lynch JB, Nat Commun, 14, 3669, 2023), did the authors perform correlation studies between Turicibacter and bile acids?
Response 3: Thanks for sharing the literature to give us a deeper understanding. According to our correlation results between Turicibacter and bile acids, there was no significant correlation. In the article you mentioned, the researchers cultured individual strains and performed bile transformations in vitro. They also find that colonization with individual Turicibacter strains leads to changes in mice bile acid profiles. This excellent study reveals how Turicibacter modifies host bile acids and lipid metabolism. Our analysis did not find a correlation between Turicibacter and bile acids in the present study. Hence, we did not mention Turicibacter data in Figure 7D. This difference may be caused by confounding factors such as mouse breed, feeding conditions, and disease model. The correlation data between Turicibacter and bile acids is presented in the attachment.
Comments 4: Suppl. Table 2: Details of SOCS1 (used for western blotting), CD4+, CD45+ (used for IF staining) and IL1b, IL6, TNFa and COX2 (used for IHC staining) are missing.
Response 4: Thank you for pointing this out. We have now reorganized the supplemental files and listed the antibody information in the Supplement Table S2 and S3.
Comments 5: Figure legends 2 and 3- Please remove *P<0.05
Response 5: There has been a change in Figure 2, we have removed *P<0.05 in revised Figure legends 3.
Comments 6: Figure 9 (proposed model)- Replace colon ‘cavity’ with ‘lumen’
Response 6: Thank you for your comments. We have replaced colon ‘cavity’ with ‘lumen’ in revised Figure 9.
Comments 7: It is ‘Feces’ not ‘Fece’
Response 7: Thank you for correcting the grammar. In view of the ‘feces’ may cause ambiguity like ‘Comment 8’, we have replaced ‘feces’ with ‘colonic contents’.
Comments 8: It is not clear whether Feces (fecal pellets) as mentioned in Figure 8C, D or colonic contents (as mentioned in Methods) were utilized to measure BAs and SCFAs
Response 8: Thank you for pointing it out. Due to the possibility of contamination of feces, all samples for BAs and SCFAs detections were tested using colonic contents (as mentioned in Methods). To address this comment, we have accordingly revised Figure 8 to emphasized the sample type.
Comments 9: Please provide a rationale as to why only female C57BL/6J mice were used for this study?
Response 9: During the study design phase, we found that for DSS-induced colitis models, though the proportion of male mice was high, female mice were also reported in the literature. As also observed for many autoimmune diseases in humans, sex-specific differences in susceptibility to experimental colitis exist and should be taken into consideration. For example, male mice tend to be more susceptible to DSS induction than female mice (Nature Protocols, Wirtz, Stefan, 2017, DOI: 10.1038/nprot.2017.044). Generally, male mice achieved a higher model-constructed success rate and a more apparent therapeutic effect. Therefore, experiments were performed in female mice to reduce false positive findings. Additionally, we confirmed colitis in all the mice under DSS treatment based on symptoms and pathological section images, which confirmed the model of colitis. In terms of strain, C57BL/6 is the most widely used mouse inbred strain and shows intermediate to high predisposition to acute DSS colitis. DSS concentrations of 2–3% (wt/vol) in the drinking water for seven days induce marked colitis in C57BL/6 mice, with low mortality rates (as compared with 2.5–5% in Balb/c mice). Consequently, female C57BL/6J mice were used for this study. To address this comment, we have also supplemented the limitations of the present study in the discussion section of the revised manuscript.
Response to Comments on the Quality of English Language
Point 1: Please carefully proof-read the manuscript for typo and grammatical errors.
Response 1: Thank you for pointing this out. We have scrutinized the manuscript to avoid typo and grammatical errors.

Round 2
Reviewer 1 Report
I believe the manuscript reaches the publish standard now.
All look good
Reviewer 2 Report
I am satisfied with the revisions that have been made by the authors.
I have no comments.
Reviewer 3 Report
The authors have successfully replied to all the queries and concerns raised by the reviewers.
The revised manuscript is greatly improved.
Nothing is required at this time.